# Insect Meal Mixture as Sustainable Fishmeal Substitute in Rainbow Trout (*Oncorhynchus mykiss*) Diets

**DOI:** 10.3390/ani15182661

**Published:** 2025-09-11

**Authors:** Houda Abdallah, Sara Bellezza Oddon, Ilaria Biasato, Jacopo Pio Salvatico, Ilario Ferrocino, Christophe Trespeuch, Sébastien Crépieux, Alberto Brugiapaglia, Zaira Loiotine, Maria Teresa Capucchio, Muhammad Irfan Malik, Stefano Bagatella, Mohamed Salah Azaza, Laura Gasco

**Affiliations:** 1Department of Agricultural, Forest and Food Sciences, University of Turin, L.go P. Braccini 2, 10095 Grugliasco, TO, Italy; houdaabdallah1409@gmail.com (H.A.); ilaria.biasato@unito.it (I.B.); jacopopio.salvatico@unito.it (J.P.S.); ilario.ferrocino@unito.it (I.F.); alberto.brugiapaglia@unito.it (A.B.); zaira.loiotine@unito.it (Z.L.); laura.gasco@unito.it (L.G.); 2Faculty of Sciences of Bizerte, National Institute of Marine Sciences and Technologies (INSTM), Aquaculture Laboratory LR16INSTM03, University of Carthage, 28 Rue du 2 Mars 1934, Salammbô, Tunis 2025, Tunisia; med.azaza@instm.rnrt.tn; 3Mutatec, Chemin du Mitan, 84300 Cavaillon, France; c.trespeuch@mutatec.com; 4Invers, Champ de la Croix, 63720 Saint-Ignat, France; sebastien.crepieux@invers.fr; 5Department of Veterinary Sciences, University of Turin, L.go P. Braccini 2, 10095 Grugliasco, TO, Italy; mariateresa.capucchio@unito.it (M.T.C.); muhammadirfan.malik@unito.it (M.I.M.);

**Keywords:** *Hermetia illucens*, *Tenebrio molitor*, mix of insect meal, *Oncorhynchus mykiss*

## Abstract

Insects such as the black soldier fly and the yellow mealworm are more and more frequently being studied as sustainable ingredients in fish nutrition, due to their high nutritional value and lower environmental footprint compared to traditional animal protein sources. This study evaluates the effects of six diets with different substitutions of fishmeal by insect meals (*Hermetia illucens* and *Tenebrio molitor*) used individually or in combination on various performance and physiological parameters of rainbow trout. The inclusion of these insect meals had no negative impact on fish growth, nutrient digestibility, intestinal microbiota, and histopathological features of the liver and intestine and fillet composition. While some changes in feed and fillet colour were observed, overall performance and quality parameters remained unaffected. These findings support the potential of insect meals as sustainable ingredients in aquafeeds, particularly in low-fishmeal formulations.

## 1. Introduction

The global aquaculture industry is at a critical crossroad, tasked with meeting the escalating demand for animal protein while mitigating its environmental footprint. Fishmeal (FM) is a cornerstone of aquafeeds due to its balanced amino acid profile and digestibility. In 2022, only 34% of fishmeal was derived from by-products, while the remaining share came from wild-capture fisheries [1]. Achieving the United Nations Sustainable Development Goals (SDGs), especially SDG 14 (Life Below Water) and SDG 12 (Responsible Consumption and Production), requires alternative protein sources that allow for a slow decrease in aquaculture’s dependence on marine resources. Scientists are always trying to find alternative and sustainable protein sources for fish feed, and among these sources, insect meal could be an option. Among insects, *Hermetia illucens* (HI, black soldier fly) and *Tenebrio molitor* (TM, yellow mealworm) are the main insects species used as promising protein alternatives to FM in fish diets [1,2]. Insect farming represents a transformative solution to aquaculture’s sustainability challenges. The HI larvae thrive on organic waste streams, reducing landfill burdens while requiring 80% less water than soybean cultivation [3,4]. For TM, Laroche et al. [5] reported that the eco-efficiency of TM protein extracts position them between plant- and animal-based protein sources, underscoring their potential as environmentally sustainable alternatives. Hence, many studies have focused on the impact of replacing FM with TM or HI meal. These substitutions have ranged from partial to total replacement of FM, and the effects of both full-fat and defatted insect meals on zootechnical performance have been investigated. Rainbow trout (*Oncorhynchus mykiss*), a globally significant European aquaculture species with production exceeding 800,000 metric tons [6], represents an ideal model for advancing sustainable feed research. Several studies have been conducted to evaluate the substitution of FM with insect meal in the diet of this species. For instance, Caimi et al. [7] reported that partially defatted BSF meal can be included up to 15% in low-fishmeal-based diets for rainbow trout without compromising growth performance and nutrient digestibility. In addition, Stadtlander et al. [8] demonstrated that fish growth and fillet quality were generally not affected by an almost 50% replacement of fishmeal with defatted HI meal (28% inclusion level) in the diet of rainbow trout. Chemello et al. [9] suggested that partially defatted TM meal could completely replace FM at a 20% inclusion level in commercial trout diets without negatively affecting fish performance. Furthermore, the total replacement of FM by 25% of defatted TM meal in rainbow trout juveniles’ diets had no effect on whole-body composition or digestibility coefficients [10]. Melenchón et al. [11] reported that 50% substitution of FM with HI or TM larvae meals (18% inclusion level) in rainbow trout diets demonstrated that TM improved growth performance, protein utilization, and digestive activity, while both insect meals showed comparable effects on liver cellular morphology and metabolism and exhibited functional potential through modulation of antioxidant responses and tissue health. Similarly, the inclusion of HI meal at up to 32% in the total substitution of FM in diets for rainbow trout had no negative effects on nutrient digestibility, growth performance, somatic indices, or intestinal histomorphology [12]. Additionally, beneficial modulation of the gut microbiota was observed, characterized by an increase in short-chain fatty acid (SCFA)-producing bacteria and a decrease in foodborne pathogenic bacteria. The inclusion of 15% of HI larvae meal shifted the gut community toward probiotic taxa, increasing *Lactobacillus* and *Bacillus* while reducing *Aeromonas*. Metagenomic predictions also indicated enhanced carbohydrate metabolism in HI-fed fish, supporting the positive impact of HI meal on gut health and overall performance.

Although HI and TM are the most extensively studied alternative protein sources in aquafeeds, their potential synergistic effects in rainbow trout remain largely unexplored. Therefore, this study aimed to evaluate the effects of including HI, TM, and their mixtures at different inclusion levels in rainbow trout diets. The investigation focused on key parameters such as pellet colour, which can influence feed acceptance and intake, as well as growth performance, somatic indices, fillet quality, nutrient digestibility, intestinal microbiota, and histopathological features of the liver and intestine, to provide a comprehensive understanding of the fish’s response to these insect-based diets.

## 2. Materials and Methods

The trial was conducted at the Experimental Facility of the Department of Agricultural, Forest, and Food Sciences (DISAFA) of the University of Turin, Italy. The experimental protocols for the growth and digestibility trials involving rainbow trout were designed in compliance with the guidelines outlined by current European and Italian laws governing the care and use of experimental animals (European Directive 86/609/EEC, transposed into Italian law by D. L. 116/92). The experimental protocol received approval from the Ethical Committee of the University of Turin (protocol no. 0565302 of 11 October 2023).

### 2.1. Insect Meal and Experimental Diets

Insect meals (HI and TM) were provided by MUTATEC (Cavaillon, France) and INVERS (Saint-Ignat, France), respectively. The larvae of the two insect species were reared on plant-based by-products. In terms of processing, HI larvae were subjected to a mechanical process for partial defatting, while TM larvae were not subjected to any treatment, except drying. According to the nutritional requirements of rainbow trout [13], seven isonitrogenous (50%, as fed), lipidic (22%, as fed), and energetic (23.5 MJ/kg, as fed) diets were formulated and extruded by Sparos LDA (Olhão, Portugal). A control diet (CTRL) was formulated to mimic a commercial formulation containing a low level of FM (15%), with the remaining protein provided by plant sources. The six experimental diets were designed with different inclusion levels of HI, TM, and a combination of both (HI and TM) while maintaining the same crude protein, ether extract, and gross energy contents. Two diets, HI100 and TM100, involved fully replacing FM in the CTRL diet with HI meal and TM meal, respectively. Finally, four diets included increasing levels of a 1:1 mixture of HI and TM to replace 25% (MIX25), 50% (MIX50), 75% (MIX75), or 100% (MIX100) of the FM in the CTRL diet. Table 1 displays the ingredients in the seven diets. The chemical composition, essential amino acids (EAAs), non-essential amino acids (NEAAs), and other components of the experimental diets, analyzed by the Sparos company, are presented in Appendix A. To evaluate nutrient digestibility, 1% of indigestible marker (Celite^®^; Fluka, St. Gallen, Switzerland) was added to all diets. To prepare the experimental extruded diets, the powdered ingredients were mixed according to the target formulation. Subsequently, the mixture was processed into pellets with a diameter of 2.0 mm using a twin-screw extruder. After extrusion, the pellets were dried and cooled. Once coated with oils through vacuum coating, the pellets were packaged in sealed plastic buckets and transported to the research site.

### 2.2. Feed Colour Evaluation

A representative sample (about 250 g) of feed for each treatment was distributed in shallow trays, and colour was evaluated at five points on the surface using a Chroma Meter CR-400 (Konica Minolta Sensing Inc., Osaka, Japan), with results expressed in terms of lightness (*L**), redness (*a**), and yellowness (*b**) in the CIELAB colour space model [14].

### 2.3. Growth Trial

#### 2.3.1. Fish and Experimental Conditions

The fish were purchased from the private fish hatchery “Fattoria del Pesce” (Marano Ticino [NO], Italy). The total duration of the experiments was 99 days, with the first 15 days constituting the acclimatization period of the animals to the experimental conditions and the remaining 84 days for data collection. Three hundred and seventy-eight rainbow trout were individually weighed (126.2 ± 1.7 g, initial body weight [IBW]) using a precision scale (KERN PLE-N v. 2.2; KERN & Sohn GmbH, Balingen-Frommern, Germany; d: 0.1) and randomly divided into 21 fibreglass 400 L tanks with an effective water column of about 330 L (three replicate tanks per diet, eighteen fish per tank, and an initial biomass density of about 7 kg/m^3^). Feed was distributed by hand twice a day, at 9:00 and 15:00. The daily feed ration, initially set at 1.45% of the tank biomass, was gradually reduced and adjusted based on fish growth during the trial until 1.32% of the tank biomass was reached. To update the daily feed quantity, the biomass of the tanks was collectively weighed every 14 days. During feed administration, intake was checked and feed distribution was immediately interrupted if the fish stopped eating. Mortality was monitored daily. Throughout the experiment, rainbow trout were reared under controlled conditions with a water flow rate of 8 L/min per tank, a temperature of 14 ± 1 °C, and dissolved oxygen levels maintained at 8.85 ± 0.36 mg/L. Temperature and dissolved oxygen were monitored in situ using an oximeter (OxyGuard, Polaris C, Farum, Denmark).

#### 2.3.2. Growth Performance

At the end of the growth trial, fish were starved for 24 h, anesthetized (MS-222, PHARMAQ Ltd., Fordingbridge, Hampshire, UK; 60 mg/L), counted, and weighted. The following parameters were then calculated:(1)$$Survival\;rate\left(SR,\%\right)=\frac{number\;of\;fish\;at\;the\;end\;of\;the\;trial}{number\;of\;fish\;at\;start}\times 100$$(2)$$Weight\;gain\left(WG,g\right)=FBW\left(final\;body\;weight,g\right)-IBW\left(initial\;body\;weight,g\right)\;$$(3)$$Specific\;growth\;rate\left(SGR,\%/d\right)=\frac{\ln\left(FBW\right)-ln\left(IBW\right)}{number\;of\;feeding\;day}\times 100\;$$(4)$$Feed\;conversion\;ratio\left(FCR\right)=\frac{total\;feed\;supplied\left(g,DM\right)\;}{WG\left(g\right)}$$(5)$$Protein\;efficiency\;ratio\left(PER\right)=\frac{WG\left(g\right)}{dry\;protein\;intake\left(g\right)}$$

#### 2.3.3. Condition Factor, Carcass Yield, and Somatic Indices

A total of 126 fish (6 fish per tank, 18 fish per treatment) were randomly selected, sacrificed by overdose of anesthetic (MS-222, PHARMAQ Ltd., Fordingbridge, Hampshire, UK; 500 mg/L), and individually measured to evaluate the condition factor (K). Fish were weighted and eviscerated to calculate carcass yield (CY), somatic indices (hepatosomatic index, HSI, and viscerosomatic index, VSI), and the fatness coefficient (CF), as detailed below:(6)$$K=\frac{fish\;weight\left(g\right)}{body\;length\;\;{\left(cm\right)}^{3}}\times 100$$(7)$$CY\left(\%\right)=\frac{total\;weight\;without\;gut\;and\;gonad\left(g\right)}{fish\;weight\left(g\right)}\times 100$$(8)$$HSI\left(\%\right)=\frac{liver\;weight\left(g\right)}{fish\;weight\left(g\right)}\times 100$$(9)$$VSI\left(\%\right)=\frac{gut\;weight\left(g\right)}{fish\;weight\left(g\right)}\times 100$$(10)$$CF\left(\%\right)=\frac{perivisveral\;fat\;weight\left(g\right)}{fish\;weight\left(g\right)}\times 100$$

### 2.4. Sampling and Analysis

#### 2.4.1. Histomorphology

At the end of the trial, one portion of the anterior intestine, liver, and spleen was dissected from five fish per tank (fifteen fish/diet) and fixed into 10% formalin solution. Selected sections were subsequently processed and dehydrated through a graded series of ethanol (70%, 80%, 95%, and 100%), clarified in isoparaffin, and embedded in paraffin wax following standard histological protocols. The paraffin-embedded blocks were cut at 5 μm thickness with a microtome (RM2245, Leica, Milan, Italy), and the obtained tissue sections were stained with hematoxylin and eosin (H&E) for histomorphometric evaluations. Histological slides were examined with a photonic microscope (Zeiss Axiophot, Oberkochen, Germany), photographed with a CMOS Discovery C30 camera, and processed with Fiji [15]. Slides were evaluated for morphometric measurements of the following parameters: villus height (VH, from the tip of the villus to the base), villus width (at the apex, middle portion, and bottom of the villus), villus area (VA), and mucosal, submucosal, and muscularis thicknesses, which were assessed on ten well-oriented and intact villi. The surface area of the villi was calculated using a formula by Sakamoto et al. [16]:(11)$$VA\;({\mathrm{mm}}^{2})=\left[\left(2\pi \right)\right.\times (VW/2)\times (VH)]$$
where VW = villus width and VH = villus height. The mucosal, submucosal, and muscularis thicknesses were also measured at 5 standardized points of the gut layers for each captured field. In addition, the following histopathological alterations were evaluated: gut inflammation, inflammation and white pulp hyperplasia/depletion in the spleen, and inflammation/degeneration in the liver. All observed histopathological alterations were evaluated using a semiquantitative scoring system as follows: absent (score = 0), mild (score = 1), moderate (score = 2), or severe (score = 3). Inflammatory infiltrates were also assessed considering type and pattern as follows: absent (score = 0), mononuclear (score = 1), mixed with neutrophils (score = 2), or mixed with eosinophils (score = 3) for type, and absent (score = 0), focal (score = 1), multifocal (score = 2), disseminated (score = 3), or diffuse (score 4) for pattern.

#### 2.4.2. Gut Microbiota

At the end of the growth trial and after assessments and sampling for evaluations at Section 2.3.2, Section 2.3.3 and Section 2.4.1, the fish were fed one day before dissection to ensure their intestines were full. One hundred and five fish (five fish per each tank, fifteen fish per treatment) were sacrificed by an overdose of anesthetic (MS-222, PHARMAQ Ltd., Fordingbridge, Hampshire, UK; 500 mg/L). The intestines were carefully dissected, and pressure was delicately applied from the anterior to the distal end to collect the intestinal contents. The samples were transferred into 2 mL sterile Eppendorf tubes and stored at −80 °C until DNA extraction and sequencing. DNA extraction was carried out using the DNeasy PowerSoil Pro Kit^®^ (Thermo Fisher Scientific, Waltham, MA, USA) following the manufacturer’s instructions. DNA concentrations were measured with the Qubit Flex Fluorometer using the dsDNA High-Sensitivity (HS) Assay Kit (Invitrogen, Carlsbad, CA, USA, Thermo Fisher Scientific). For gut content samples (500 mg starting material), DNA was extracted using the RNeasy Power Microbiome Kit (Qiagen, Milan, Italy) according to the manufacturer’s instructions. The extracted DNA was quantified using a NanoDrop ND-1000 spectrophotometer (Thermo Scientific, Milan, Italy) and adjusted to a final concentration of 10 ng/µL. The PCR solution was prepared with 25 µL of extracted DNA, Taq polymerase, 16S Primer F, 16S Primer R, and water. A thermal cycler was used to amplify the V3–V4 region of the 16S rRNA gene following an optimized protocol. The resulting PCR amplicons were purified according to the Illumina metagenomic standard procedure. Sequencing was performed on an Illumina platform, generating high-quality paired-end reads for downstream microbiota analysis.

#### 2.4.3. Fillet Quality

For each treatment, fifteen fish were filleted (five fish per tank, fifteen fish per treatment). The left fillet was immediately frozen for chemical analyses (Section 2.6), while the right-side fillets were weighed, sealed in plastic bags, and stored at 4 °C. After 24 h, fillets were gently blotted with paper to eliminate surface moisture and then weighed to calculate drip loss (DL).(12)$$DL\left(\%\right)=\frac{\left[raw\;fillet\;weight\left(g\right)-raw\;fillet\;weight\;after\;24\;h\left(g\right)\right]}{raw\;fillet\;weight\left(g\right)}\times 100$$

The muscle pH at 24 h and colour were measured using a Crison MicropH 2001 pH meter (Crison Instruments, Barcelona, Spain) and a bench Chroma Meter CR-400 colorimeter (Konica Minolta Sensing Inc., Osaka, Japan), respectively. Results were expressed in terms of *L**, *a**, and *b** in the CIELAB colour space model [14].

The fillets were then individually vacuum-sealed in plastic bags and stored at −20 °C. Once completely frozen, they were thawed at +4 °C, removed from the packaging, gently dried with paper, and weighed to determine thawing loss (TL) using the following calculation:(13)$$TL\left(\%\right)=\frac{\left[raw\;fillet\;weight\left(g\right)-thawed\;fillet\;weight\left(g\right)\right]}{raw\;fillet\;weight\left(g\right)}\times 100$$

The same fillets were then vacuum sealed in plastic bags and cooked in a bain-marie with a water temperature of 80 °C (ED; JULABO, Seelbach, Germany) until the core temperature of the fillets reached 75 °C. The probe for measuring the internal temperature (HD 9219; Delta OHM, Caselle Di Selvazzano, Italy) was inserted into the fillet with the heaviest weight to ensure that the desired temperature was reached at the core. The fillets, on average, were cooked for about 7 min. When the core temperature reached 75 °C, the bags were removed from the bain-marie and cooled in fresh water for 10 min to stop the cooking process. The fillets were then removed from the bags, gently dried with paper, and weighed to determine cooking loss (CL) using the following formula:(14)$$CL\left(\%\right)=\frac{\left[thawed\;fillet\;weight\left(g\right)-cooked\;fillet\;weight\left(g\right)\right]}{thawed\;fillet\;weight\left(g\right)}\times 100$$

After the cooking loss determination, cooked fish samples (2.5 cm × 2.5 cm) were used to determine the texture profile analysis. The sample was collected from the cranial part of the fillets, near the lateral line. Assessment was conducted following the procedures described by Monteiro et al. [17] using a TAHD plus Texture Analyzer (Stable Micro Systems Ltd., Godalming, UK). The parameters measured were the hardness (N), the springiness, the adhesiveness (N*s), the cohesiveness, the gumminess (N), the chewiness (N), and the resilience. Some of the parameters reported for this analysis are non-dimensional because they are derived from ratios of areas or distances.

### 2.5. Digestibility Trial

The digestibility experiment used the indirect acid-insoluble ash method to calculate the apparent digestibility coefficients (ADCs) of the seven experimental diets. One hundred and five fish from the same batch as used for the growing trial, with an average initial body weight of 127 ± 0.37 g, were used in the experiment, and they were put into seven 250 L cylindroconical tanks (one per treatment, fifteen fish per tank) supplied with water in a flow-through open system. Fish were hand-fed twice a day, at 9:00 and 15:00, seven days a week. After a 10-day acclimatization period to the experimental diets, feces were collected twice daily from each tank for three weeks in a row using a continuous automatic device (Choubert’ system) [18]. To achieve the required number of replicates per treatment (*n* = 3), the experiment was carried out in three sequential blocks using the same batch of fish. Prior to each new fecal collection phase, a 10-day acclimation period was applied to allow fish to adapt to the new diets [19], and the seven treatments were always given to different tanks. The collected feces were freeze-dried and analyzed as reported in Section 2.6. The ADCs of dry matter (DM), crude protein (CP), and ether extract (EE) were calculated according to Bureau et al. [20].

### 2.6. Chemical Analyses of Insect Meal, Diets, Feces, and Fillets

The chemical analyses of diets, feces, and fillets were conducted at the DISAFA laboratory (Grugliasco, Italy). Feed samples were grounded using a cutting mill (MLI 204; Bühler AG, Uzwil, Switzerland) and analyzed following AOAC International methods [21] for DM (AOAC#934.01), CP (AOAC#984.13; nitrogen to protein conversion factors: 6.25 for feeds), and ash (AOAC#942.05). The EE of the diets, feces, and fillets was determined following AOAC #2003.05 [22], while for the insect meals, this method was slightly modified by doubling the extraction time. Gross energy (GE) content was measured using an adiabatic bomb calorimeter (C7000; IKA, Staufen, Germany). The chitin content of the diets was determined according to Woods et al. [23]. The method of analysis involved a degreasing treatment, followed by a demineralization step with hydrochloric acid and a deproteinization step with sodium hydroxide. The amino acid content of the diets, as well as the taurine, total phosphorus, EPA, and DHA content (Appendix A), was provided by the feed manufacturers. Frozen left fillets (five fish per tank, fifteen fish per treatment) were filleted, freeze-dried (Edwards MF 1000, Milan, Italy), and finely grounded (Grindomix GM200; Retsch GmbH, Haan, Germany).

### 2.7. Statistical Analysis

The assumption of normality was checked using the Shapiro–Wilk test. After assessing the normality or non-normality of the distribution of the dependent variables, the assumption of equal variances was tested using Levene’s test for homogeneity of variances. If this assumption was not met, the Brown–Forsythe statistic was applied. Pairwise multiple comparisons were performed to test the difference between each pair of means using the following post hoc tests: Tukey’s test or Tamhane’s T2 (for equal or unequal variances, respectively, under one-way ANOVA) or Dunn’s test (under the Kruskal–Wallis test). One-way ANOVA or Kruskal–Wallis tests were used to compare data (feed colour; growth performance; carcass yield; somatic indices; nutrient digestibility coefficients; fillet quality; and chemical analyses of insect meal, diets, feces and fillets) among the experimental groups. For all parameters, the replicate was considered based on the number of samples or analyses per treatment, and the fixed factor for all variables was the seven treatments. Analysis was carried out using IBM SPSS Statistics v. 27.0 for Windows. For the gut morphometric measurements, data were analyzed by fitting a generalized linear model (GLMM, lme4 R package), in which the diet was included as a fixed factor. The statistical analysis was carried out by using R (version 4.4.0). Histomorphometry scores were analyzed by fitting a cumulative link mixed-effects model (CLMM, ORDINAL R package). The results were expressed as the mean and pooled standard error of the mean (SEM). The level of significance was considered set at *p* < 0.05. For the analysis of intestinal microbiota, QIIME2’s diversity script was used to calculate alpha diversity indices. Bray–Curtis dissimilarity was used to assess differences in microbial community composition among treatments compared to the control diet.

## 3. Results

### 3.1. Pellet Colour

Colour indices (*L**, *a**, and *b**) were significantly influenced by the diets (*p* < 0.001) (Table 2). *L** values were highest in the CTRL and MIX25 diets, while HI100 and MIX75 were the lowest. For the *a** index, CTRL and TM100 diets had the highest values, followed by MIX25, which did not differ significantly (*p* > 0.05). HI100 showed the lowest *a** value, with MIX50, MIX75, and MIX100 falling in between. In terms of *b** values, CTRL and MIX25 again had the highest readings, with HI100 at the lowest end. TM100 and MIX50 showed similar, intermediate values (*p* > 0.05), while the rest also fell into the intermediate range.

### 3.2. Growth Performance

The survival rate (SR), growth performance, and feed utilization of fish fed the seven experimental diets are presented in Table 3. The MIX75-fed fish were characterized by lower SR when compared to MIX25, TM100, HI100, and the control (*p* < 0.05). All the other performance parameters were not affected by dietary treatments (*p* > 0.05).

### 3.3. Condition Factor, Carcass Yield, and Somatic Indices

The results for condition factor, carcass yield, and somatic indices indicate that there were no statistically significant differences observed for any of the calculated indices across the various experimental diets (*p* > 0.05) (Table 4).

### 3.4. Histomorphometry

The height, width, and surface area of the villus were not affected by dietary inclusion of HI or TM meals nor by varying levels of a mixture of TM and HI (*p* > 0.05). Similarly, the thicknesses of the mucosa, submucosa, and muscularis layer were also not influenced (*p* > 0.05) (Table 5). The histopathological scores related to inflammation in the gut, liver, and spleen (pattern, type, and severity) were also not significantly different (*p* > 0.05) (Table 6).

### 3.5. Gut Microbiota

Gut microbiota analysis was performed on six dietary treatments (CTRL, HI100, TM100, MIX25, MIX50, and MIX75) to evaluate the effects of dietary ingredient substitution. The MIX100 treatment was excluded due to a technical issue encountered during DNA extraction, which resulted in insufficient DNA quality and quantity for sequencing. Despite repeated attempts, the extraction could not be successfully replicated, making it impossible to include this group in the microbiota analysis. Consequently, no microbiota data are available for the MIX100 group. No significant differences were observed in alpha diversity indices (Shannon, observed species, Chao1; *p* < 0.05) (Table 7), and PCoA of beta diversity showed no significant group differences (*p* > 0.05) (Figure 1). Microbiota characterization revealed seven phyla, 10 classes, 17 orders, 27 families, and 53 genera (ASV database). The community was mainly composed of Firmicutes (30–92%), Proteobacteria (0.25–48%), and Bacteroidota (0.03–11.91%). These percentages were calculated based on the assumption that the total relative abundance of all identified phyla summed to 100%. Other phyla were detected at lower levels, depending on the treatment, such as Actinobacteria (5.20% in HI100), Cyanobacteria (1.04% in MIX75), Verrucomicrobiota (3.14% in TM100), and Desulfobacterota (6.96% in CTRL) (Figure 2). Other phyla appeared at lower levels: Actinobacteria (5.20% in HI100), Cyanobacteria (1.04% in MIX75), Verrucomicrobiota (3.14% in TM100), and Desulfobacterota (6.96% in CTRL) (Figure 2). Despite no alpha diversity differences (*p* > 0.05), ASV relative abundance varied across treatments (Figure 3). Nevertheless, the relative abundance of amplicon sequence variants (ASVs) revealed variations in the metataxonomic composition among the different dietary treatments (Figure 3), with several bacterial genera distributed across multiple phyla, although no statistically significant differences in alpha diversity were observed (*p* > 0.05). Within Firmicutes, *Mycoplasmoidaceae* was more frequent in the CTRL group, while the HI100, TM100, and MIX25 diets showed higher levels of *Metamycoplasmataceae*, *Ligilactobacillus*, *Blautia*, and *Lachnospira*. Proteobacteria (*Aeromonas*) were abundant in the TM100, HI100, and MIX25 groups, less abundant in the MIX50 group, and absent in the CTRL and MIX75 groups. *Enterococcus* was identified in the HI100 and MIX75 groups, while *Enterobacteriaceae* were more abundant in the MIX25 and MIX50 groups compared to the other treatments. *Deefgea* was more abundant in the TM100 and MIX25 groups and less abundant for the HI100, MIX50, and MIX75 treatments. Within the phylum Bacteroidota, *Phocaeicola* were identified in the HI100, TM100, MIX25, and MIX75 groups. *Weissella* was found in all groups, while *Ruminococcus* appeared only in fish fed insect meal-based diets. Finally, the phylum Desulfobacterota included *Mailhella*, which was present in the CTRL and other treatments.

### 3.6. Physical Characteristics and Proximate Composition of Fillets

The flesh colour, 24 h pH, and water holding capacity of rainbow trout fillets were influenced by the experimental diets (Table 8). Only the *L** parameter was significantly affected by the diets, with MIX75 showing a lower value when compared to TM100, MIX25, and MIX100 (*p* < 0.01). No significant difference was observed for *a** and *b** (*p* > 0.05). Fillet pH_24h_ varied significantly, with MIX25 showing a lower value than MIX50, MIX75, and CTRL (*p* < 0.001). The drip loss was higher in MIX75 when compared to HI100 (*p* < 0.001), while no differences were observed for TL or CL (*p* > 0.05) (Table 8). The texture profile analysis (Table 9) showed no significant differences among treatments (*p* > 0.05). Similarly, the proximate composition of fillets remained unaffected by the experimental diets (*p* > 0.05) (Table 10).

**Figure 3 animals-15-02661-f003:**
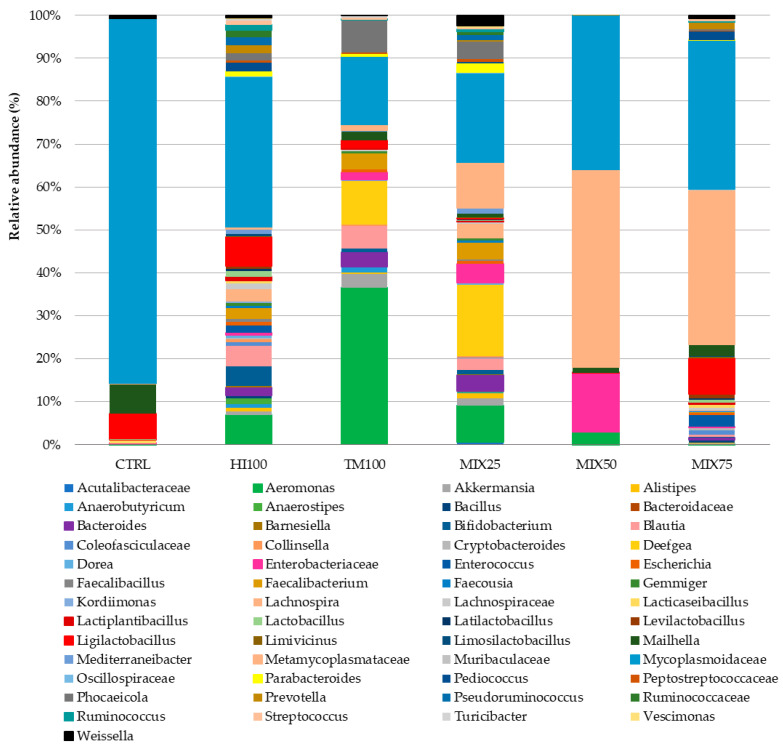
Mean relative abundance (%) of the bacterial community at genus levels in intestinal content of trout fed graded levels of a mixture of *Hermetia illucens* and *Tenebrio molitor* larvae meals.

### 3.7. Digestibility Trial

The statistical analysis of ADC_DM_, ADC_CP_, and ADC_EE_ revealed no significant differences among the dietary treatments (*p* > 0.05) (Table 11).

## 4. Discussion

Pellet colour influences feed intake in fish, especially in visually oriented species, by enhancing visibility, attractiveness, and potentially growth performance, and it can indicate ingredient composition and processing quality [24]. The *L**, *a**, and *b** values of all diets showed significant variation when compared to the control diet. The MIX25 diet was similar to the control, while the other diets had lower *L**, *a**, and *b** values, making them darker. The control diet without insect meal was lighter than all others except MIX25. The colour of insect meal greatly depends on the production process, especially the drying temperature [25]. For example, high-temperature processing can promote the formation of dark-coloured compounds, mainly through the Maillard reaction, a nonenzymatic reaction between sugars and proteins that produces coloured and aromatic compounds [25,26,27]. Enzymatic browning, on the other hand, can be attributed in part to phenolic compounds released from the cuticle or tegument of the insect. As a result, the product could undergo oxidation, protein–polyphenol interaction, and a reaction with phenoloxidase, which catalyzes the browning of the insect meals [28,29]. Microwave drying yields lighter meals, whereas defatting and extrusion can also darken colour depending on parameters like screw speed and temperature [25,30]. Increased insect meal inclusion did not consistently affect CIELAB parameters, and colour differences may result from extrusion or the interaction of HI and TM meals. No detailed CIELAB colour analysis exists in the literature for insect-based fish feeds, making comparisons difficult; Kari et al. [31] assessed pellet colour visually, which is not directly comparable. After an 84-day feeding trial, survival differed significantly, with the highest mortality in MIX75 compared to TM100, HI100, MIX25, and CTRL. As the maximum inclusion levels were 18.6% HI (HI100), 21.85% TM (TM100), and 20.3% (10.15% HI + 10.15% TM) in MIX100, it is unlikely that insect meals or their combination caused this result, since these levels are below those previously reported to induce stress in trout. Growth performance (FBW, WG, SGR, FCR, PER) was not negatively affected by total FM replacement with HI, TM, or their combination, confirming previous findings: up to 50% FM replacement (15% or 21% HI inclusion) [7,32], 100% substitution (32% HI) [12], and 25–100% FM replacement (5–20% TM) [9] showed no adverse effects. Variability among studies likely reflects differences in insect meal origin, processing, fat content, fish species, age, and rearing conditions. The observed stability of growth performance is reflected in the unaltered physiological conditions and somatic indices. The present findings demonstrate that HI, TM, and their combination can completely replace FM (up to 100%) without negatively affecting K, CY, HSI, VSI, and CF. All treatments yielded K values above 1, indicative of a good physiological condition and comparable to values previously reported in rainbow trout fed HI [12,32] and TM meals [9]. De Francesco et al. [33] argued that different K values could be associated with different fat synthesis and deposition processes due to different feeding regimes in trout. No significant variation in K and CF was observed, suggesting that insect meal inclusion did not alter lipid metabolism or deposition patterns. Similarly, carcass yield and somatic indices were not significantly affected, consistent with previous observations in rainbow trout fed HI [12,32] and TM (up to a 28% inclusion level) [34]. The only exception was a significantly higher HSI in fish fed the TM100 diet (*p* < 0.05) [9]. The intestine is directly involved in nutrient digestion and absorption and serves as a protective barrier against microbes gaining access to the gut. It is well known that dietary factors may positively or negatively influence intestinal health or, specifically, villus morphology [35]. In the present study, no significant differences were observed in anterior gut morphometry among treatments, in agreement with Caimi et al. [7] (15% partially defatted HI, 50% FM replacement). Similarly, TM inclusion (10.5% HI + 10.5% TM, up to 100% FM replacement) did not alter intestinal morphometry, consistent with Melenchón et al. [11] and Józefiak et al. [36]. The histomorphometric parameters associated with the use of TM have been little investigated in the literature and, apart from two studies by Józefiak et al. [36] and Melenchón et al. [11], most have not reported changes in these parameters [37,38,39]. Regarding histopathological alterations of the anterior gut, liver, and spleen, they varied from absent to moderate in all the organs. The absence of adverse effects related to dietary insect meal inclusion observed in the organs examined in the present study is in agreement with the available literature [3,7,11,40]. The unaffected growth performance indicates that the HI and TM meals supplied a level of nutritional quality comparable to fishmeal. Moreover, intestinal morphometry and histopathology of the anterior gut, liver, and spleen remained unchanged, confirming the digestive safety and physiological compatibility of insect meals at 100% FM replacement (10.5% HI + 10.5% TM). The replacement of fishmeal with insect-based protein sources, such as HI and TM, in rainbow trout diets induced genus-level shifts in the gut microbiota that reflect adaptive responses to novel dietary substrates like chitin, lipids, and amino acids. The gut microbiota data from six experimental treatments (CTRL, HI100, TM100, MIX25, MIX50, MIX75) assessing the effects of fishmeal substitution were included in the analysis. The dietary inclusion of HI, TM, or their combinations in rainbow trout did not induce significant changes in gut microbial richness, as indicated by alpha diversity (Shannon, observed species, and Chao1; *p* > 0.05) or beta diversity indices. Nevertheless, a trend toward increased microbial diversity was observed in fish fed insect-based diets. These results align with previous findings in rainbow trout, where the administration of HI-containing diets resulted in either comparable or elevated alpha diversity, as measured by the Chao1 and Shannon indices, relative to the control group [41,42]. Firmicutes, Proteobacteria, and Bacteroidota dominated all treatments, in line with previous studies [43]. *Mycoplasmoidaceae* were detected in the intestines of trout across all dietary treatments, HI100, TM100, MIX25, MIX50, and MIX75, and they were dominant in the CTRL group, while *Metamycoplasmataceae* was more abundant in the HI100, TM100, and MIX25 treatments [44]. *Mycoplasma* is considered a gut symbiont that contributes to host metabolism by utilizing dietary substrates to produce lactic and acetic acids as the main fermentation products [45]. While these species are often associated with pathogenicity, their persistence here may indicate a commensal role in nutrient scavenging, particularly in degrading chitin, a structural polysaccharide abundant in insect exoskeletons. A high abundance of *Mycoplasmoidaceae* in the gut microbiota of juvenile rainbow trout has also been previously reported in salmonids [41]. The abundance of *Ligilactobacillus* and *Lachnospira*, known for producing short-chain fatty acids (SCFAs) like acetate and butyrate, in the gut microbiota for fish fed different diets (HI100, TM100, and MIX25 diets) underscores their probiotic role in maintaining gut barrier integrity and modulating immune responses [46]. These genera likely benefit from the chitin-rich composition of insect meals, as SCFA production in carnivorous fish has been linked to increased dietary fibre availability, particularly from indigestible components such as chitin, rather than dietary lipid content [12,43]. These SCFA-producing genera are indicators of a healthy gut in fish. Their existence in control groups suggests stable microbial functionality, whereas their reduction in other treatments (MIX50 and MIX75) may reflect dietary or environmental stressors [47]. The genus *Aeromonas* was observed significantly more frequently in the TM100, HI100, and MIX25 groups, less so in the MIX50 group, and it was absent in the CTRL and MIX75 groups. This genus may act as a commensal or opportunistic pathogen, depending on the context. Its presence in low abundance is generally considered normal in healthy fish, whereas overgrowth may indicate a microbiota imbalance or stress-related factors [48]. In this study, the gut microbiota showed a positive shift toward SCFA-producing bacteria and reduced foodborne pathogens, suggesting a beneficial increase in diversity associated with the improved ability of fish to cope with intestinal disorders [12]. Regarding fillet physical quality, fillet colour is a key parameter influencing freshness perception and consumer choice [49]. In this study, only the *L** parameter of the CIELAB colour space was affected by the experimental diets. The *L** value observed for the CTRL diet was comparable to those of all other treatments. Given the similarity between HI100 and TM100, it can be inferred that the inclusion of HI and TM meals does not have a significant impact on fillet colour. Similar findings were reported by Secci et al. [50] and Caimi et al. [7] with HI (full-fat or partially defatted) and by Iaconisi et al. [51] for up to 50% full-fat TM meals in *O. mykiss*. In blackspot sea bream (*Pagellus bogaraveo*), a 40% inclusion of full-fat TM meal significantly increased the *b** value [52], possibly due to fatty acids or pigments such as riboflavin [53] and β-carotene [54], which are known to influence the *a** value when more than 50% of FM is replaced by TM or HI meals. For pH_24h_, significant differences were observed—the lowest in MIX25 and the highest in MIX75—while most insect diets showed lower values than the CTRL diet. This trend may be attributed to the modified buffering capacity and protein composition introduced by the inclusion of HI and TM meals. The relatively higher pH in MIX75 suggests a potential synergistic effect at this inclusion level, possibly stabilizing post-processing acidity. In contrast to the present study, Caimi et al. [7] found no effect of HI, and Iaconisi et al. [51] reported no effect with up to 50% TM on the pH_24h_ value of rainbow trout fillets. Regarding WHC, only the DL values were influenced by the experimental treatments, with MIX75 showing a significant difference compared to the CTRL diet. For TL and CL, the different experimental treatments did not affect the values obtained. Similar results were reported by Secci et al. [50] for the CL (up to 50% full-fat HI), Caimi et al. [7] for the DL, Avramiuc [55] for the TL, and Iaconisi et al. [51] (up to 50% TM) for rainbow trout fillets. Regarding the TPA, the use of the experimental diets to feed *O. mykiss* caused no differences in the parameters evaluated. Melenchón et al. [56] reported no changes in the TPA parameters of fish fillets with the inclusion of insect meals in the diet. Similarly, Chaklader et al. [57] found that full-fat and defatted HI meals did not affect adhesiveness or springiness but increased fillet hardness, chewiness, and gumminess, potentially reducing fillet quality. Additionally, studies on shear force, such as those by Caimi et al. [7] and Iaconisi et al. [51], indicated no significant changes in rainbow trout fillets with the inclusion of insect meals. Finally, the experimental diets were designed to be isonitrogenous, isolipidic, and isoenergetic and they did not affect fillet chemical composition, confirming efficient nutrient utilization. Similar results were observed by Dumas et al. [58], Rema et al. [10], and Caimi et al. [7] for HI meal-based diets and Jeong et al. [34] and Iaconisi et al. [52] for TM meal-based diets. The apparent digestibility of nutrients in the six treatment diets was similar to that recorded for the CTRL diet, as previously presented in earlier studies [7,12]. In this study, the different inclusion levels of insect meal mixtures of HI and TM in trout did not have a negative effect on nutrient digestibility. For all treatments, the ADCDM was around 80%, while the apparent digestibility of nutrients such as crude protein (ADCCP) and ether extract (ADCEE) was around 95%. These high digestibility values across all diets suggest that the feeds provided sufficient nutrient availability, which is consistent with the similar growth performance observed among all treatment groups. Similarly, the graded substitution levels of FM with TM meal (20%, 30%, 60%, and 100%), corresponding to 5%, 7.5%, 15%, and 25% inclusion levels, respectively, had no effect on the apparent digestibility coefficients of dry matter, protein, fat, phosphorus, and energy in trout [10]. In contrast, several studies have reported that high inclusion levels of insect meal can negatively affect the apparent digestibility [59,60,61]. This reduction in digestibility is generally attributed to the increased chitin content and the lower digestibility of certain insect-derived components at higher concentrations, which may impair nutrient absorption and gastrointestinal function. For example, Liland et al. [59] reported that in salmon, rainbow trout, sea bass, or seabream, the digestibility of crude protein begins to decrease at an insect meal inclusion level of 25%. This decrease is often associated with an increase in chitin (above 2–3%), which negatively affects the specific growth rate and feed conversion ratio [62].

## 5. Conclusions

The findings of this study suggest that partially defatted HI meals, full-fat TM meals, and a mixture of both can be considered suitable ingredients in low fish meal-based diets for rainbow trout. While pellet colour revealed significant differences between treatments, dietary insect meal utilization had no adverse effects on growth performance, somatic indices, or fillet chemical composition. Moreover, the analysis of nutrient and energy digestibility coefficients, the intestinal morphometry and histopathology of the anterior gut, liver, and spleen, and the intestinal microbiota analysis showed no significant differences between the treatments, including mixtures of HI and TM meals. No negative effects were observed on the overall physical quality of the fillets, although slight variations were noted in colour lightness (*L**), pH, and water holding capacity (drip loss, DL). Other evaluated parameters, such as fillet texture, were not affected. In conclusion, the inclusion of HI and TM meals, whether individually or in combination, did not negatively influence the physical quality of rainbow trout fillets, and their combined use further highlights the potential for synergistic effects, as mixtures performed comparably to or, in some cases, slightly better than single-meal inclusions.

## Figures and Tables

**Figure 1 animals-15-02661-f001:**
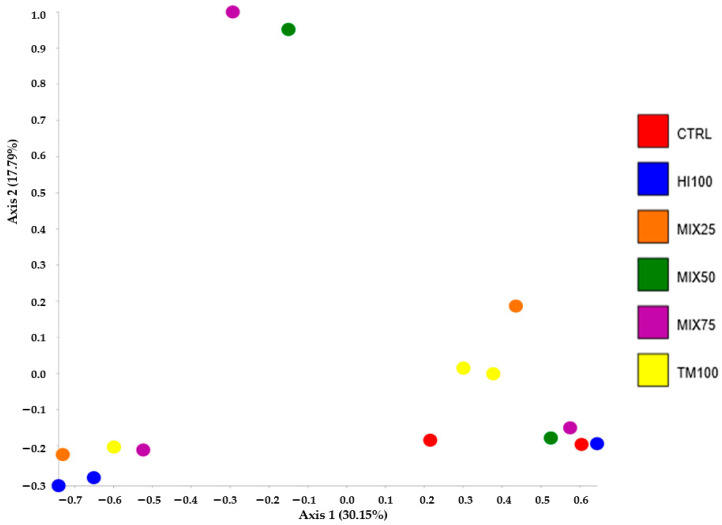
Distribution of bacteria with beta analysis (PCoA plot, Emperor) of seven diets of rainbow trout. Each point represents one replicate. PCoA, Principal Coordinate Analysis.

**Figure 2 animals-15-02661-f002:**
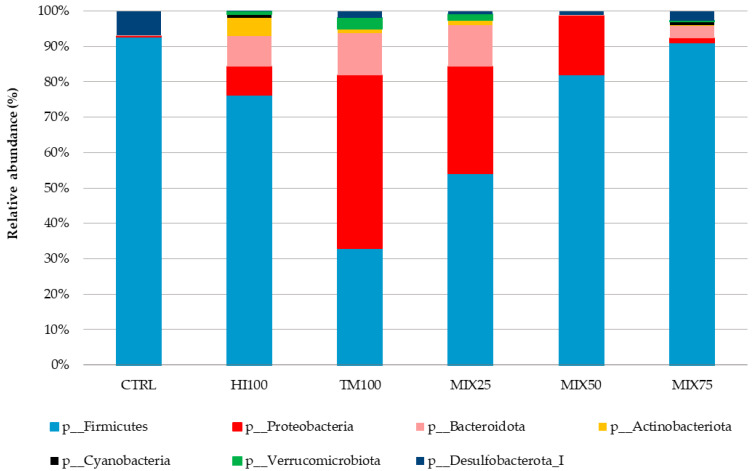
Mean relative abundance (%) of the bacterial community at phyla levels in the intestinal content of trout fed graded levels of a mixture of *Hermetia illucens* and *Tenebrio molitor* larvae meals.

**Table 1 animals-15-02661-t001:** Ingredients of rainbow trout diets (g/100 g, as feed).

Ingredients	CTRL	HI100	TM100	MIX25	MIX50	MIX75	MIX100
Fishmeal LT70	15.00			11.25	7.50	3.75	
HI		18.60		2.45	5.05	7.60	10.15
TM			21.85	2.45	5.05	7.60	10.15
Soy protein concentrate	18.00	18.00	18.00	18.00	18.00	18.00	18.00
Wheat gluten	11.00	11.00	11.00	11.00	11.00	11.00	11.00
Wheat meal	9.00	9.00	9.00	9.00	9.00	9.00	9.00
Poultry meal	7.00	7.00	7.00	7.00	7.00	7.00	7.00
Wheat starch	5.40	0.83	0.22	4.30	2.94	1.62	0.43
Corn gluten meal	5.00	5.00	5.00	5.00	5.00	5.00	5.00
Soybean meal	5.00	5.00	5.00	5.00	5.00	5.00	5.00
Fish protein hydrolysate	2.00	2.00	2.00	2.00	2.00	2.00	2.00
VMP	1.00	1.00	1.00	1.00	1.00	1.00	1.00
Choline chloride 50%	0.20	0.20	0.20	0.20	0.20	0.20	0.20
Antioxidant	0.20	0.20	0.20	0.20	0.20	0.20	0.20
Sodium propionate	0.10	0.10	0.10	0.10	0.10	0.10	0.10
Monoammonium phosphate	1.20	2.40	2.30	1.50	1.80	2.05	2.30
Celite	1.00	1.00	1.00	1.00	1.00	1.00	1.00
L-Lysine HCl 99%	0.20	0.60	0.45	0.30	0.37	0.45	0.50
DL-Methionine		0.17	0.18	0.05	0.09	0.13	0.17
Fish oil	6.20	7.00	7.00	6.40	6.60	6.80	7.00
Rapeseed oil	12.50	10.90	8.50	11.80	11.10	10.50	9.80

Abbreviations: CTRL, control diet; HI, *Hermetia illucens* meal; TM, *Tenebrio molitor* meal; HI100, 100% of fish meal replaced by HI meal; TM100, 100% of fish meal replaced by TM meal; MIX25, MIX50, MIX75, and MIX100, respectively, with 25%, 50%, 75%, and 100% of fish meal replaced by a mixture (1:1) of HI and TM meals; VMP, vitamin and mineral premix.

**Table 2 animals-15-02661-t002:** Pellet colour of the experimental diets tested in rainbow trout. The values are average values (*n* = 4).

Items	Experimental Diets							SEM	*p*-Value
	CTRL	HI100	TM100	MIX25	MIX50	MIX75	MIX100		
*L**	34.21 ^a^	27.12 ^c^	30.79 ^b^	34.83 ^a^	30.94 ^b^	27.81 ^c^	29.92 ^b^	0.533	<0.001
*a**	7.04 ^a^	4.82 ^c^	6.93 ^a^	6.35 ^ab^	5.93 ^b^	5.89 ^b^	5.94 ^b^	0.144	<0.001
*b**	26.43 ^a^	18.27 ^e^	22.93 ^b^	25.26 ^a^	22.43 ^b^	19.78 ^d^	21.02 ^c^	0.524	<0.001

Abbreviations: CTRL, control diet; HI, *Hermetia illucens* meal; TM, *Tenebrio molitor* meal; HI100, 100% of fish meal replaced by HI meal; TM100, 100% of fish meal replaced by TM meal; MIX25, MIX50, MIX75, and MIX100, respectively, with 25%, 50%, 75%, and 100% of fish meal replaced by a mixture (1:1) of HI and TM meals; (L), lightness; (a), redness; (b), yellowness; SEM, standard error of the mean; different superscripts in a row indicate significant differences (a, b, c, d, e); *n*, replicates.

**Table 3 animals-15-02661-t003:** Survival rate and growth performances of rainbow trout fed the experimental diets (*n* = 3).

Items	Experimental Diets							SEM	*p*-Value
	CTRL	HI100	TM100	MIX25	MIX50	MIX75	MIX100		
SR (%)	96.30 ^ab^	92.59 ^ab^	94.44 ^ab^	100 ^a^	87.04 ^bc^	81.48 ^c^	87.04 ^bc^	1.653	0.047
IBW (g)	128.93	135.50	154.77	154.63	123.00	128.70	140.97	4.173	0.239
FBW (g)	254.30	262.27	281.30	281.43	249.60	254.00	266.73	4.268	0.236
WG (g)	125.36	126.27	126.56	126.77	126.60	125.29	125.76	0.374	0.894
SGR (%.day^−1^)	0.81	0.79	0.73	0.72	0.85	0.81	0.76	0.016	0.317
FCR	1.36	1.27	1.26	1.19	1.26	1.35	1.31	0.034	0.902
PER	1.47	1.55	1.60	1.68	1.61	1.48	1.53	0.038	0.827

Abbreviations: CTRL, control diet; HI, *Hermetia illucens* meal; TM, *Tenebrio molitor* meal; HI100, 100% of fish meal replaced by HI meal; TM100, 100% of fish meal replaced by TM meal; MIX25, MIX50, MIX75, and MIX100, respectively, with 25%, 50%, 75%, and 100% of fish meal replaced by a mixture (1:1) of HI and TM meals; IBW, initial body weight; FBW, final body weight; WG, weight gain; SGR: specific growth ratio; SR, survival ratio; FCR, feed conversion ratio; PER, protein efficiency ratio; different superscripts in a row indicate significant differences (a, b, c); *n*, replicates.

**Table 4 animals-15-02661-t004:** Condition factor, carcass yield, somatic indices, and coefficient of fatness of rainbow trout fed the experimental diets (*n* = 18).

Items	Experimental Diets							SEM	*p*-Value
	CTRL	HI100	TM100	MIX25	MIX50	MIX75	MIX100		
K	1.11	1.16	1.17	1.16	1.14	1.15	1.12	0.008	0.234
CY (%)	90.16	89.83	89.95	89.06	89.92	90.07	89.45	0.134	0.298
HSI (%)	1.21	1.19	1.17	1.18	1.27	1.21	1.27	0.014	0.390
VSI (%)	8.55	8.24	8.44	9.21	8.25	8.24	8.62	0.115	0.314
CF (%)	2.69	2.88	2.97	2.90	2.99	2.70	2.85	0.084	0.943

Abbreviations: CTRL, control diet; HI, *Hermetia illucens* meal; TM, *Tenebrio molitor* meal; HI100, 100% of fish meal replaced by HI meal; TM100, 100% of fish meal replaced by TM meal; MIX25, MIX50, MIX75, and MIX100, respectively, with 25%, 50%, 75%, and 100% of fish meal replaced by a mixture (1:1) of HI and TM meals; K, condition factor; CY, carcass yield; HSI, hepatosomatic index; VSI, viscerosomatic index; CF, coefficient of fatness; *n*, number of fish.

**Table 5 animals-15-02661-t005:** Anterior gut morphometric indexes of rainbow trout fed the experimental diets (*n* = 15).

Parameters	Experimental Diets							SEM	*p*-Value
	CTRL	HI100	TM100	MIX25	MIX50	MIX75	MIX100		
VH (mm)	0.587	0.627	0.628	0.636	0.571	0.615	0.598	0.068	0.634
WH (mm)	0.139	0.150	0.156	0.147	0.140	0.150	0.144	0.066	0.549
VA (mm^2^)	0.259	0.301	0.313	0.296	0.255	0.293	0.271	0.109	0.383
Gut mucosal thickness (mm)	0.653	0.704	0.718	0.741	0.675	0.634	0.701	0.078	0.439
Gut submucosal thickness (mm)	0.122	0.112	0.120	0.117	0.120	0.112	0.115	0.648	0.768
Gut muscularis thickness (mm)	0.164	0.155	0.159	0.161	0.156	0.186	0.174	0.109	0.622

Abbreviations: CTRL, control diet; HI, *Hermetia illucens* meal; TM, *Tenebrio molitor* meal; HI100, 100% of fish meal replaced by HI meal; TM100, 100% of fish meal replaced by TM meal; MIX25, MIX50, MIX75, and MIX100, respectively, with 25%, 50%, 75%, and 100% of fish meal replaced by a mixture (1:1) of HI and TM meals; VH, villus height; VW, villus width; VA, villus surface area; *n*, number of fish.

**Table 6 animals-15-02661-t006:** Pathological scoring of the anterior gut, liver, and spleen of rainbow trout fed the fed experimental diets (*n* = 15).

Parameters	Experimental Diets							SEM	*p*-Value
	CTRL	HI100	TM100	MIX25	MIX50	MIX75	MIX100		
Anterior gut mucosa									
Inflammation pattern		2.214	2.466	2.285	2.466	2.333	1.900	0.746	0.626
Inflammation type	2.000	1.714	2.066	1.785	1.857	2.000	1.733	0.099	0.215
Inflammation severity	1.500	1.035	1.200	1.214	1.678	1.533	1.200	0.647	0.234
Anterior gut submucosa									
Inflammation pattern	2.266	1.785	2.200	1.785	1.785	2.400	2.133	0.876	0.355
Inflammation type	2.000	1.714	2.000	1.714	1.714	2.000	1.866	0.102	0.333
Inflammation severity	1.000	0.750	0.900	0.714	0.928	1.200	0.966	0.651	0.202
Anterior gut muscularis									
Inflammation pattern	0.266	0.142	0.533	0.285	0.500	0.666	0.466	1.014	0.669
Inflammation type	0.266	0.142	0.533	0.428	0.571	0.666	0.533	1.015	0.706
Inflammation severity	0.066	0.035	0.133	0.107	0.142	0.200	0.133	1.015	0.676
Liver									
Inflammation pattern	0.466	0.785	0.357	0.714	0.533	1.000	0.600	0.800	0.548
Inflammation type	0.266	0.571	0.214	0.428	0.266	0.533	0.400	0.803	0.571
Inflammation severity	0.133	0.214	0.178	0.214	0.166	0.366	0.200	0.801	0.495
Degeneration	0.433	0.607	0.571	0.285	0.733	0.333	0.366	0.688	0.875
Spleen									
Inflammation pattern	0.266	0.000	0.153	0.142	0.133	0.133	0.133	0.125	0.912
Inflammation type	0.133	0.000	0.076	0.071	0.066	0.066	0.133	0.072	0.916
Inflammation severity	0.066	0.000	0.038	0.035	0.066	0.066	0.066	0.037	0.869
Hyperplasia	0.366	0.133	0.192	0.285	0.266	0.400	0.300	0.740	0.600
Depletion	0.033	0.000	0.153	0.000	0.000	0.033	0.033	0.036	0.787

Abbreviations: CTRL, control diet; HI, *Hermetia illucens* meal; TM, *Tenebrio molitor* meal; HI100, 100% of fish meal replaced by HI meal; TM100, 100% of fish meal replaced by TM meal; MIX25, MIX50, MIX75, and MIX100, respectively, with 25%, 50%, 75%, and 100% of fish meal replaced by a mixture (1:1) of HI and TM meals; *n*, number of fish.

**Table 7 animals-15-02661-t007:** Shannon_entropy, observed features, and ace of rainbow trout fed the fed experimental diets (*n* = 3).

Items	Experimental Diets						SEM	*p*-Value
	CTRL	HI100	TM100	MIX25	MIX50	MIX75		
Shannon_entropy	6.47	8.81	8.01	8.95	6.00	6.98	0.464	0.414
Observed features	202.00	1256.00	662.00	1104.50	159.50	510.33	188.577	0.543
ace	209.77	1414.58	710.54	1212.87	163.65	565.48	213.226	0.532

Abbreviations: CTRL, control diet; HI, *Hermetia illucens* meal; TM, *Tenebrio molitor* meal; HI100, 100% of fish meal replaced by HI meal; TM100, 100% of fish meal replaced by TM meal; MIX25, MIX50, and MIX75, respectively, with 25%, 50%, 75%, and 100% of fish meal replaced by a mixture (1:1) of HI and TM meals; *n*, replicates.

**Table 8 animals-15-02661-t008:** Physical quality of fillets from rainbow trout fed experimental diets (*n* = 15).

Items	Experimental Diets							SEM	*p*-Value
	CTRL	HI100	TM100	MIX25	MIX50	MIX75	MIX100		
Colour and pH after 24 h									
*L^*^*	45.27 ^ab^	44.81 ^ab^	46.22 ^a^	46.69 ^a^	45.73 ^ab^	43.94 ^b^	46.22 ^a^	0.212	0.004
*a^*^*	3.65	3.51	3.99	3.30	3.19	4.06	3.04	0.105	0.053
*b^*^*	5.13	6.22	6.04	6.23	5.74	5.67	5.86	0.106	0.073
pH_24h_	6.33 ^bc^	6.23 ^ab^	6.22 ^ab^	6.18 ^a^	6.30 ^bc^	6.39 ^c^	6.24 ^ab^	0.012	<0.001
Water holding capacity									
DL (%)	3.96 ^ab^	3.62 ^a^	4.44 ^bc^	3.80 ^ab^	4.53 ^bc^	5.05 ^c^	4.41 ^bc^	0.098	0.001
TL (%)	7.29	6.63	7.33	7.26	6.93	7.25	6.49	0.163	0.722
CL (%)	8.68	8.57	8.97	9.62	8.36	7.90	9.17	0.191	0.330

Abbreviations: CTRL, control diet; HI, *Hermetia illucens* meal; TM, *Tenebrio molitor* meal; HI100, 100% of fish meal replaced by HI meal; TM100, 100% of fish meal replaced by TM meal; MIX25, MIX50, MIX75, and MIX100, respectively, with 25%, 50%, 75%, and 100% of fish meal replaced by a mixture (1:1) of HI and TM meals; *L**, lightness; *a**, redness; *b**, yellowness; DL, drip loss; TL, thawing loss; CL, cooking loss; different superscripts in a row indicate significant differences (a, b, c); *n*, number of fish.

**Table 9 animals-15-02661-t009:** Effects of the experimental diets on the texture profile analysis of the rainbow trout fillets (*n* = 15).

Items	Experimental Diets							SEM	*p*-Value
	CTRL	HI100	TM100	MIX25	MIX50	MIX75	MIX100		
Hardness (N)	3.933	3.825	3.482	3.751	3.280	3.667	3.465	0.083	0.387
Springiness	0.658	0.619	0.650	0.617	0.643	0.671	0.639	0.006	0.225
Adhesiveness (N*s)	−0.005	−0.007	−0.003	−0.005	−0.004	−0.004	−0.003	0.0004	0.093
Cohesiveness	0.447	0.442	0.460	0.430	0.426	0.440	0.440	0.00	0.640
Gumminess (N)	1.780	1.697	1.626	1.640	1.417	1.603	1.525	0.047	0.567
Chewiness (N)	1.175	1.060	1.070	1.015	0.915	1.071	0.980	0.034	0.579
Resilience	0.155	0.149	0.164	0.150	0.146	0.149	0.153	0.002	0.503

Abbreviations: CTRL, control diet; HI, *Hermetia illucens* meal; TM, *Tenebrio molitor* meal; HI100, 100% of fish meal replaced by HI meal; TM100, 100% of fish meal replaced by TM meal; MIX25, MIX50, MIX75, and MIX100, respectively, with 25%, 50%, 75%, and 100% of fish meal replaced by a mixture (1:1) of HI and TM meals; *n*, number of fish.

**Table 10 animals-15-02661-t010:** Proximate composition fillets (g/100 g ww) of rainbow trout fed experimental diets (*n* = 12).

Items	Experimental Diets							SEM	*p*-Value
	CTRL	HI100	TM100	MIX25	MIX50	MIX75	MIX100		
DM	24.67	24.98	24.96	25.21	24.32	24.91	24.85	0.122	0.633
CP	20.46	20.62	20.15	20.23	20.55	20.11	20.16	0.060	0.093
EE	4.08	3.94	4.33	4.52	3.47	4.52	4.10	0.121	0.271
Ash	1.42	1.42	1.41	1.44	1.41	1.40	1.45	0.007	0.456

Abbreviations: CTRL, control diet; HI, *Hermetia illucens* meal; TM, *Tenebrio molitor* meal; HI100, 100% of fish meal replaced by HI meal; TM100, 100% of fish meal replaced by TM meal; MIX25, MIX50, MIX75, and MIX100, respectively, with 25%, 50%, 75%, and 100% of fish meal replaced by a mixture (1:1) of HI and TM meals; ww, wet weight; *n*, number of fish.

**Table 11 animals-15-02661-t011:** Apparent digestibility coefficients (%) of dry matter, proteins, and ether extracts of rainbow trout fed the experimental diets (*n* = 3).

Items	Experimental Diets							SEM	*p*-Value
	CTRL	HI100	TM100	MIX25	MIX50	MIX75	MIX100		
ADC_DM_ (%)	82.32	82.76	85.96	83.73	83.90	83.02	86.65	0.463	0.083
ADC_CP_ (%)	95.90	94.80	95.83	96.13	96.10	95.14	95.91	0.151	0.103
ADC_EE_ (%)	97.38	97.46	98.05	97.63	98.14	97.91	98.19	0.115	0.337

Abbreviations: CTRL, control diet; HI, *Hermetia illucens* meal; TM, *Tenebrio molitor* meal; HI100, 100% of fish meal replaced by HI meal; TM100, 100% of fish meal replaced by TM meal; MIX25, MIX50, MIX75, and MIX100, respectively, with 25%, 50%, 75%, and 100% of fish meal replaced by a mixture (1:1) of HI and TM meals; ADC_DM_ (%), apparent digestibility coefficient of dry matter; ADC_CP_ (%), apparent digestibility coefficient of protein; ADC_EE_ (%), apparent digestibility coefficient of ether extract (lipid); *n*, replicates.

## Data Availability

Data from this study are available from the corresponding authors upon reasonable request.

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
